# Evidence of emotion dysregulation as a core symptom of adult ADHD: A systematic review

**Ana-María Soler-Gutiérrez** [1,2], **Juan-Carlos Pérez-González** [3], **Julia Mayas** [1]*

**1** Faculty of Psychology, Universidad Nacional de Educación a Distancia (UNED), Madrid, Spain, **2** Escuela Internacional de Doctorado de la UNED (EIDUNED), Universidad Nacional de Educación a Distancia (UNED), Madrid, Spain, **3** Faculty of Education, Universidad Nacional de Educación a Distancia (UNED), Madrid, Spain

* jmayas@psi.uned.es

## Abstract

Attention Deficit Hyperactivity Disorder (ADHD) is a developmental disorder, with an onset in childhood, that accompanies the person throughout their life, with prevalence between 3 and 5% in adults. Recent studies point towards a fourth core symptom of the disorder related to the emotional information processing that would explain the repercussions that ADHD has on the social, academic, and professional life of the people affected. This review aims to describe emotion dysregulation features as well as the brain activity associated in adults with ADHD. A search of the scientific literature was launched in specialized data-bases: PsycInfo, Medline, Eric, PsycArticle, Psicodoc and Scopus, following PRISMA guidelines. Twenty-two articles met the inclusion criteria: (a) an ADHD clinical diagnosis, (b) participants over 18 years old, (c) emotion regulation measurement, (d) empirical studies, and (c) in English. Due to the heterogeneity of the studies included, they were classified into three sections: measures and features of emotion regulation (ER) in people with ADHD, neurological and psychophysiological activity related to ER, and treatments. The studies found that meet the selection criteria are scarce and very heterogeneous both in aims and in sample features. Adults with ADHD show a more frequent use of non-adaptive emotion regulation strategies compared to people without ADHD symptoms. Moreover, emotion dysregulation was associated with symptom severity, executive functioning, psychiatric comorbidities, and even with criminal conviction. Different patterns of brain activity were observed when people with and without ADHD were compared. These results may suggest that psychopharmacological treatments as well as behavioral therapies could be useful tools for improving emotional difficulties in adult ADHD.

## Introduction

Attention Deficit Hyperactivity Disorder (ADHD) is one of the most prevalent neurodevelopmental disorders (between 5–7% among children and adolescents and between 3–5% among the adult population [1]) with onset in childhood that can persist into adulthood. It is characterized by a persistent pattern of three cardinal symptoms: inattention, impulsiveness and / or

**Data Availability Statement:** All relevant data are within the paper and its Supporting Information files.

**Funding:** This work was supported by Universidad Nacional de Educación a Distancia (UNED) through

an FPI-UNED grant given to A.M.S.G. The funders had no role in the study design, data collection and analysis, the decision to publish, or the preparation of the manuscript.

**Competing interests:** The authors have declared that no competing interests exist.

hyperactivity describing three specific subtypes: combined, predominantly inattentive and predominantly hyperactive-impulsive [2]. According to DSM-5 [2] criteria for diagnosis of ADHD, symptoms must appear before the age of 12, with a duration of at least 6 months in two or more contexts that negatively impact social and academic/occupational activities. ADHD is described as a disorder with very heterogeneous manifestations [3], of multifactorial etiology [4] and with different manifestations throughout lifespan [5]. The diagnosis of ADHD in adults is much more difficult because of the lack of specific criteria and the high comorbidity with other disorders [6].

The diagnosis of ADHD continues to be strictly clinical and based exclusively on the behavioral symptoms of inattention, impulsivity and hyperactivity, despite the recent accumulation of evidence suggesting other affective difficulties frequently associated with the disorder beyond simple comorbidity [7]. For example, through a longitudinal network model, a recent study by Karalunas et al. [8] has associated temperamental irritability with ADHD symptomatology, supporting the hypothesis that this type of anger dysregulation might be a frequent feature of ADHD. Likewise, this study also confirmed that working memory (i.e., as an indicator of executive functioning) had independent direct influences on both ADHD and depression.

It is well established that ADHD is associated with executive function (EF) deficits [9]. According to Barkley [10], ADHD is both, a self-regulating deficit disorder and an executive function disorder, since affected people have greater difficulties using EF to self-regulate and achieve their goals. Other studies, however, have pointed out anatomical-functional alterations of the reward system with motivational deficits being a primary dysfunction of ADHD, as well as noting that the last deficits are independent of the executive ones [11]. In recent years, new evidence has emerged that suggests that emotion regulation deficits, or emotion dysregulation (ED), are a core symptom of ADHD [12, 13].

Neuroscience research has identified neural networks related to EF [14], reward system [15], and emotional information processing [16–18]. Moreover, an overlap has been found between EF and emotional neural networks [19–21]. Interestingly, neuroimaging studies have found that the brain structures involved in these processes are affected in ADHD, including the dorsal anterior cingulated cortex and posterior cingulated cortex [22], dorsolateral prefrontal cortex, ventrolateral prefrontal cortex, parietal cortex, striatum, and cerebellum [23, 24], and the amygdale [25, 26]. Finally, other studies have focused on social impairments in ADHD that correlate with a reduced volume of subcortical areas: insula, amygdala and striatum [27].

Nigg and Casey [28] have proposed a model based on three neural networks that would be affected in ADHD and that would be at the basis of emotional, motivational, and cognitive deficit. The fronto-striatal network, which would account for the difficulties in working memory and response selection; the fronto-cerebellar network, involved in temporal organizing and timing of behavior; and finally, the fronto-limbic pathway that would explain the emotional regulation deficit. Specifically, the orbitofrontal cortex (OFC) modulates the limbic system activation and integrates the emotional information that comes from this and other brain areas [19, 28]. Its abundant interconnection with the dorsolateral prefrontal cortex (dlPFC), results in the modulation and control of emotional responses, adapting them to personal goals and the social context [29, 30]. Thus, evidence may indicate that ADHD is associated with an inadequate top-down regulation of emotional reactivity by higher cortical regions [10, 12].

## Emotion regulation in ADHD

In recent years, emotion regulation (ER) has become a focus of interest in the study of the emotional difficulties observed in ADHD [12, 31]. According to Gross [32], emotional self-

regulation or ER can be defined as a complex process by which people modulate their emotions to direct their behavior towards goals using strategies (for example, cognitive change strategy of reappraisal and response modulation strategy of suppression, see [33, 34]), which start when there are overlearned responses that conflict with the desired goal [35]. Thus, executive functions and emotion regulation might be closely linked and interrelated [10]. As Bailey and Jones [108] claim, emotional regulation, along with cognitive and social regulation, would develop and become increasingly complex, based on the essential components of inhibition, attention, working memory and shifting. In fact, some studies suggest that appropriate EF skills are associated with the inhibition of negative affect expressions [36], as well as with a greater use of the reappraisal strategy [37]. On the contrary, the habitual use of the suppression strategy is related to difficulties in daily EF [37].

Emotional dysregulation appears very early in ADHD childhood compared to children with typical neurodevelopment [38]. Furthermore, there is abundant research on childhood ADHD that finds associations between emotion dysregulation and symptom severity (for reviews see [39, 40]), failures in emotion recognition [41–43], working memory deficits [6, 12, 44], poor parental ER skills [45–48], even with abnormal functioning of the autonomic nervous system [49, 50].

In contrast to the vast evidence on ED in children with ADHD, little is known about ED in adults with ADHD. The differences in aspects related to ER in people with and without ADHD are not conclusive. For example, while some studies find that adults with ADHD use the maladaptive ER strategy of suppressing emotions more frequently than controls [51, 52], others do not find this [53].

Contradictions are also observed depending on the emotional assessment instrument used in the different studies (for instance, [54, 55]). One variable that could explain the contradictory results is the high comorbidity present in adults with ADHD, which makes it difficult to assess emotional aspects independently [6, 56]. More studies are needed to clarify what emotional aspects are impaired in adult ADHD, what factors are involved, as well as what tools are the most appropriate to assess them. Meanwhile, this review aims to locate and unify the most relevant results in ED obtained so far in adults diagnosed with ADHD.

## Aims

The main goal of this work was to conduct a systematic review (SR) of the literature on emotional dysregulation in adults with ADHD.

Moreover, three specific goals are proposed: (1) to analyze emotion dysregulation in adults with ADHD; (2) to examine ADHD symptoms and their relationship with ED; (3) to identify ADHD neuro-functional abnormalities related to ED. In line with these specific goals, we hypothesize that adults with ADHD will score lower on ER measurements, associated with greater impairment (H1); will use maladaptive emotion regulation strategies (H2), and will have a different ER-related pattern of brain activity (H3) than controls.

## Method

A systematic literature review (SLR) was carried out, following the principles and phases of the PRISMA model [57]. Although a meta-analytical study would be the most desirable, the systematic review methodology seems to be the most appropriate in this case due to the small number and the heterogeneity of the selected articles.

### Inclusion and exclusion criteria

The studies had to meet the following criteria to be included: (a) participants diagnosed with ADHD following DSM-IV or later criteria, (b) over 18 years old, (c) to have an ER measure, (d) empirical studies, (e) published in English.

The exclusion criteria were: (a) the lack of a clinical diagnosis of ADHD (e.g., symptoms or probable ADHD) or not being ADHD the main disorder of the study, (b) including people under 18 years of age in the study without offering analysis of the data by age range, (c) not providing ADHD participants ER measures (e.g., parents), (d) reviews, qualitative studies or meta-analyses.

### Procedure

A search was launched through EBSCOhost in PsycInfo, Medline, Eric, PsycArticle and Psico-doc, and in Scopus databases from February 1 to 25, 2022. The following search criteria were entered into the databases anywhere to the title, keywords, or abstract fields, filtered by academic publications in English: [ADHD OR "Attention deficit hyperactivity disorder"] AND ["Emotion regulation" OR "Emotion Dysregulation"] AND [adult*]. An informal search was also carried out on Google Scholar throughout the first fortnight of March 2022. The process of eliminating non-relevant papers following PRISMA guidelines [57] can be seen in the flowchart (Fig 1). When the criteria were not clear, the inclusion or exclusion of the article was decided among all three authors. S1 Table shows the bibliometric properties of the included articles [13, 51–54, 58–74], which are marked with * in the references section.

## Results

Twenty-two studies met all the inclusion criteria. However, given the great heterogeneity of the studies selected, they were classified into three sections. The first section included studies on measures and features of ER in people with ADHD. The second section comprised studies that analyze the neurological and psychophysiological activity during the performance on ER

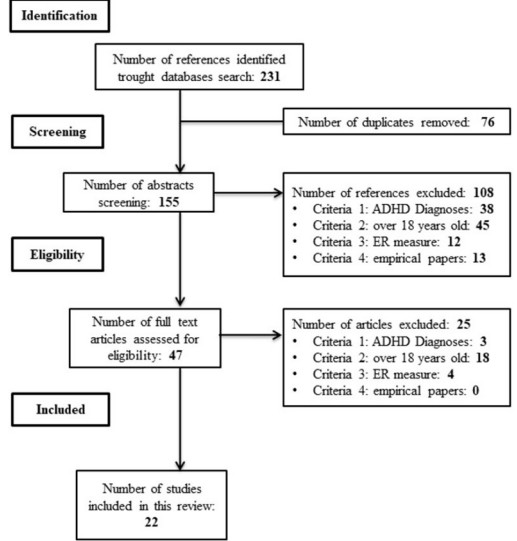

**Fig 1. PRISMA flowchart.**

tasks in ADHD. Finally, the third section included interventions that provided ER outcomes in adults with ADHD.

S2 Table shows the demographic variables of the studies selected showing a wide heterogeneity, especially in terms of sample size, gender, ADHD subtypes, comorbidity, and medication status. However, most of the ADHD sample comes from clinical contexts (see S2 Table).

Regarding the methodological variables, Table 1 summarizes the measurement tools used in each study (see also Table 2), as well as the main results obtained related to emotional regulation, which are described below. Results were presented in three different sections. Finally, Table 2 lists the assessment tools used [53, 75–97].

## Emotion regulation features in adult ADHD

Ten studies compared ER scores in adults with ADHD and controls, finding that ADHD group had consistently lower ER scores than control groups, ranging from medium to very large effect sizes [$d' = 0.31–2.27$] [51–54, 58–63] (see Table 1). In a series of two studies, Hirsch et al. [13, 60] investigated the role of ER in ADHD. In a first study, they carried out confirmatory factor analysis [13]. They found a good model fit to the ADHD four-factor model, with values below 0.10 in RMSEA [0.07; 90% confidence interval: 0.068–0.071] and around 2 in $\chi^2/df$ ratio [2.03], suggesting that ED is a core symptom of ADHD. In a second study [60], they carried out a person-centered approach, through cluster analysis, to distinguish ADHD subtypes based on the presence or absence of ED. They found a two-cluster solution [$F = 87.21$; $\eta^2 = 0.185$] which, once the variables with small effect size were eliminated, explained 25% of the variance between clusters. The differences in the classification variables were significant and large in size [specifically in the ERQS: $d = 1$, $p < 0.0001$]. They found that Cluster 2 was loaded with higher proportion of combined ADHD presentation, women, and ED. Moreover, other studies also showed that women with ADHD presented greater emotion dysregulation [64] and emotionality [65].

Some studies found that emotion dysregulation was significantly associated with impulsivity [13], executive function deficits and functional impairment [66]; as well as total Adult ADHD Self-Report Scale Symptom Checklist (ASRSv1.1) score with non-adaptative cognitive strategies [63]. High scores of emotion dysregulation were also associated with substance use disorder [64], criminal conviction [67], BPD comorbidity [62, 63] and sluggish cognitive tempo comorbidity [66], as well as recall of parental ADHD symptoms [65]. However, no significant differences were found between ADHD emotional dysregulation scores and other disorders such as autism spectrum disorder [59]. Results regarding BPD were inconclusive. Cavelti et al. [58] found no significant difference between ADHD and BPD, but when they compared them together with controls, they found significant differences [$F_{4, 179} = 0.64$, $p < 0.001$]. "Confronting emotions with self-encouragement" was the factor from ERQS that best discriminated between the clinical group and the control group [$F_{2, 180} = 5.29$, $p = 0.006$]. Moukhtarian et al. [62] found no significant differences between ADHD and BPD in emotion dysregulation measured through retrospective reports, although they found that the ADHD+BPD group reported elevated ratings compared to the only ADHD group. Nevertheless, Rüfenacht et al. [63] found that the ADHD group had higher emotion dysregulation scores compared to other clinical and community samples [$t = 4.79$; $p < 0.001$] but less than BPD and ADHD + BPD [$p < 0.001$]. As for the comparison in ED between ADHD and cyclothymia, Brancati et al. [68] only found significant differences in negative emotionality and in negative/ positive emotionality ratio measured with the Reactivity, Intensity, Polarity, and Stability questionnaire (RIPoSt-40), in which adults with cyclothymia scored higher than ADHD.

**Table 1. Methodological information and main results of the reviewed articles.**

| Author | Diagnostic Criteria | ADHD measures | ER measures | Statistical Analysis | Main Results |
|---|---|---|---|---|---|
| Anker et al. [67] | DSM-V | DIVA 2.0, ASRS-v1.1 | DESR | $t$ Test, logistic regression | DESR CC > No-CC: $t$ = 3.89*, $p$ < **0.001** <br> Association CC-DESR: OR = 1.08, $p$ < **0.001** |
| Anker et al. [64] | DSM-V | DIVA 2.0, ASRS-v1.1 | DESR | $t$ Test, logistic regression | DESR men < women: $t$ = −5.293*, $p$ < **0.001** <br> Association DUD-DESR: OR = 1.05, $p$ = **0.006** |
| Badoud et al. [72] | DSM-V | DIVA 2.0, WURS, ASRS-v1.1 | ERS | Wilcoxon test, Cohen's $d'$ | Non-significant. <br> ERS pre-post: $d' \leq$ -0.21 |
| Brancati et al. [68] | DSM-V | DIVA 2.0 | RIPoSt-40 | ANOVA | Negative emotional dysregulation ADHD vs ADHD +CYC: <br> $F$ = 1.844, $p$ = 0.161 |
| Cavelti et al. [58] | DSM-IV-TR | WRAADDS, WURS, Adult Interview*, CAARS-L:SR/O | ERSQ | MANCOVA, ANOVA | ERSQ ADHD = BPD: $F_{4,126}$ = 0.98, $p$ = .75 <br> ERSQ ADHD/BPD < C: $F_{4,179}$ = 0.64, $p$ < **.001** <br> Confronting emotions with self-encouragement (ERSQ) BPD < ADHD < C: $F_{2,180}$ = 5.29, $p$ = **0.006** |
| Edel et al. [65] | DSM-IV | WURS | EES | $t$ Test, Pearson's $r$ | Emotionality (EES) men < women: $t$ = 2.56, $p$ = **0.013** <br> Associations QRPRB-EES: r = .26-.36; $p$ < **0.05** |
| Helfer et al. [59] | DSM-V | DIVA 2.0, BAARS-IV | WRAADDS -EDS | ANOVA, Cohen's $d'$ | WRAADS-EDS ADHD/ASD > C: F = 65.56, $p$ < **0.001** <br> WRAADDS-EDS ADHD-C: $d'$ = **2.27** (95% CI: 1.74–2.80) |
| Hirsch et al. [60] | DSM-IV | DIVA 2.0, CAARS-L:SR/O | ERSQ | Cluster Analysis | ERSQ Cluster 1 > Cluster 2: Welch Test $p$ < **0.0001**, $d'$ = **1** <br> ERSQ Cluster 1 < C: $d'$ = **0.31–0.68** <br> ERSQ Cluster 2 < C: $d'$ = **1.34–1.87** |
| Hirsch et al. [13] | DSM-IV | DIVA 2.0, CAARS-L:SR/O | ERSQ | Confirmatory Factor Analysis | $\chi^2$ = 2.03 / RMSEA = .07 <br> ERSQ- Innatention (CAARS): $r$ = - 0.20 <br> ERSQ–Impulsivity (CAARS): $r$ = - 0.30 <br> ERSQ-Negative affect: $r$ = - 0.49 <br> ERSQ–Positive affect: $r$ = 0.69 |
| Li et al. [54] | DSM-IV | CAADID, ADHD-RS, CAARS-SR | BRIEF-A-EC | $t$ Test, ANOVA | EC (BRIEF) ADHD < C: $t$ = 7.054, $p$ < **0.001** <br> Central β power-ADHD EC (BRIEF): $r$ = 0.464, $p$ = **0.027** <br> Central β power-C EC (BRIEF): $r$ = 0.129, $p$ = 0.385 |
| Materna et al. [51] | DSM-IV | WURS, CAARS-L:O | ERQ, SEK-27 | ANOVA, $t$ Test | fMR—condition x group: $p$ < **0.05** <br> Reappraisal (ERQ) ADHD < C: $t$ = 2.40, $p$ = **0.020** <br> Suppression (ERQ) ADHD > C: $t$ = −3.28, $p$ = **0.002** <br> SEK-27 ADHD < C: $t$ = −8.06, $p$ < **0.001** |
| Matthies et al. [69] | DSM-IV | WURS, CAARS-L:SR | ERQ | ANOVA, $t$ Test | Feeling of being overwhelmed by emotion–Acceptance vs suppression <br> Time x strategy: $F_{2,66}$ = 7.031, $p$ < **0.050** <br> Time effect: $F_{1,33}$ = 7.691, $p$ < **0.050** <br> Between group-T3: $t_{1,34}$ = -2.71, $p$ < **0.050** |
| Mitchell et al. [61] | DSM-IV | CAADID, CAARS | DERS | ANOVA, effect size $\eta^2$ | DERS TDAH > C: $p$ ≤ **0.02**; $\eta^2$ ≥ **0.23** <br> DERS main group effect: $p$ ≤ **0.001**; $\eta^2$ ≥ **0.47** |
| Mitchell et al. [71] | DSM-IV | Childhood ADHD Symptom, Current ADHD Symptoms Scale, CAARS, CAADID. | DERS | ANOVA, Cohen's $d'$ | Group x time: $p$ = 0.002, d = 1.63 <br> Impulse control difficulties (DERS): $p$ = **0.002**, $d$ = **1.60** <br> Limited access to ER strategies (DERS): $p$ = **0.007**, d = **1.38** |
| Moukhtarian et al. [62] | DSM-V | DIVA 2.0 | WRAADDS -EDS | Kruskal-Wallis test | WRADDS EDS: $X^2$(3) = 68.34, $p$ < **0.001** <br> C < ADHD/BPD/ADHD+BPD: $p$ < **0.001** <br> ADHD < ADHD+BPD: $p$ = **0.005** |
| Reimherr et al. [73] | DSM-IV-TR | CAADID, WRAADDS | WRAADDS -EDS | ANOVA, Cohen's $d'$ | Efficacy outcome EDS (WRAADDS) <br> Atomoxetine > Placebo: <br> F = 15.10, $p$ = **0.001**, $d'$ = **0.66** |

*(Continued)*

**Table 1.** (Continued)

| Author | Diagnostic Criteria | ADHD measures | ER measures | Statistical Analysis | Main Results |
|---|---|---|---|---|---|
| Reimherr et al. [74] | DSM-IV-TR | CAADID, WURS, WRAADS | WRAADDS-EDS | ANOVA, Cohen's $d'$ | Efficacy outcome EDS (WRAADDS) OROS methylphenidate > Placebo: $F = 10.476$, $p = 0.002$, $d' = 0.70$ |
| Rüfenacht et al. [63] | DSM-V | DIVA 2.0, ASRS-v1.1 | ERS, CERQ | $t$ Test | ERS ADHD > C: $t = 8.03$; $p < 0.001$ <br> ERS ADHD < BPD/ADHD+BPD: $t = 10.46–14.64$, $p < 0.001$ <br> CERQ Non-Adap ADHD > C: $t = 9.12–14.19$, $p < 0.001$ <br> CERQ Non-Adap ADHD < BPD/ADHD+BPD: $t = 4.04–6.62$, $p < 0.01$ |
| Shushakova et al. [52] | DSM-V | DIVA 2.0, ASRS-v1.1, ADHD-SR, WURS | ERQ, SEK-27 | ANOVA, $t$ Test, Pearson's $r$ | Reappraisal (ERQ) ADHD < C: $t_{76} = 2.60$, $p = 0.011$, $d' = 0.59$ <br> Suppression (ERQ) ADHD > C: $t_{76} = 2.96$, $p = 0.008$, $d' = 0.67$ <br> SEK-27 ADHD < C: $t_{76} = 7.59$, $p < 0.001$, $d' = 1.72$ <br> Suppression—LPP reduced: $F_{1, 69} = 7.645$, $p = 0.007$ <br> LPP ADHD (unmedicated) > C: $F_{1, 57} = 8.095$, $p = 0.006$, $d' = 0.77$ |
| Shushakova et al. [70] | DSM-V | DIVA 2.0, ASRS-v1.1, ADHD-SR, WURS | SEK-27 | ANOVA, $t$ Test, Pearson's $r$ | TDAH: ERP N2pc–SEK-27: $r = 0.35$, $p = .033$ <br> C: ERP N2 pc–SEK-27: $r = -0.106$, $p = .511$ <br> $z = -1.999$, $p = .023$ |
| Silverstein et al. [66] | DSM-V | ACDS v1.2, AISRS | AISRS-ED | t Test, Spearman $r$ | ED (AISRS) ADHD < ADHD+SCT: $t = 2.30$, $p = 0.23$, $d' = 0.53$ <br> Association ED–EFD severity: $r = 0.30$, $p < 0.01$ |
| Thorell et al. [53] | DSM-V | DIVA 2.0, Childhood ADHD Symptom, WURS, ASRS | CERI | ANOVA, Cohen's $d$ | CERI subscales ADHD > C: F = 8.9–203.66, $p < 0.05–0.001$, $d' = 0.37–1.66$ <br> No-significant differences: suppression and fear reactivity |

* Calculated from the mean and standard deviation.

Notes: CC = criminal conviction, DUD = drug use disorder, OR = odds ratio, CYC = cyclothymic disorder, BPD = borderline personality disorder, QRPRB = Questionnaire of Recalled Parental Rearing Behavior, PCA = principal components analysis, CAARS-L: large version, SR = Self-report, O = Observer-rating, fMR = functional magnetic resonance, C = control group, ANOVA = Analysis of Variance, MANCOVA = Multivariate Analysis of Covariance, LPP = late positive potential. EFD: executive functioning deficits, SCT = sluggish cognitive tempo.

ADHD and emotion regulation assessment instruments can be found in Table 2.

Additionally, an ADHD trend to use maladaptive emotional regulation strategies and to rate low their own regulatory skills measured by *Selbsteinschätzung emotionaler Kompetenzen* (SEK-27) was observed. In this regard, Materna et al. [51] found that the ADHD group scored lower on reappraisal but higher on suppression than the control group [$t = 2.40$, $p = 0.020$; $t = -3.28$, $p = 0.002$; respectively] and valued their own regulation strategies more negatively that controls did [$t = -8.06$, p < 0.001]. Similar results were obtained by Shushakova et al. [52] [$t = 2.60$, $p = 0.01$, for reappraisal; $t = 2.96$, $p = 0.008$, for suppression; $t = 7.59$, $p < 0.001$, for SEK-27] and by Rüfenacht et al. [63] [$t = 9.12–14.19$, p < 0.001, for non-adaptive Cognitive Emotion Regulation Questionnaire (CERQ) scales].

Mitchell et al. [61] analyzed the effect on ER of abstinence in smokers with and without ADHD. They found significant differences between the ADHD group and the control group [$F = 36.17$, $p < 0.001$, $\eta^2 = 0.49$]. These differences were maintained throughout the two experimental conditions (abstinence—non-abstinence), both in the Difficulties in Emotion Regulation Scale (DERS) [$p \leq 0.001$, $\eta^2 \geq 0.47$], and in the laboratory tasks [$p < 0.001$, $\eta^2 \geq 0.31$]. Increases in emotion dysregulation in the abstinence condition were due to significant

**Table 2. Assessment tools used in the articles reviewed.**

| ADULT ADHD | | | EMOTION REGULATION | | |
|---|---|---|---|---|---|
| TITLE | ABREV. | AUTHORS | TITLE | ABREV. | AUTHORS |
| *Conners Adult ADHD Diagnostic Interview for DSM-IV* | CAADID | Epstein et al. [75] | *Difficulties in Emotion Regulation Scale* | DERS | Gratz and Roemer [88] |
| *Diagnostic Interview for ADHD in adults* | DIVA 2.0 | Kooij and Francken [76] | *Emotion Regulation Questionnaire* | ERQ | Gross and John [89] |
| *ADHD Rating Scale-IV* | ADHD-RS | DuPaul et al. [77] | *Cognitive Emotion Regulation Questionnaire* | CERQ | Garnefsky et al. [90] |
| *Adult ADHD Self-Report Scale. Symptom Checklist* | ASRS-v1.1 | Kessler et al. [78] | *Emotion Reactivity Scale* | ERS | Nock et al. [91] |
| *Barkley Adult ADHD Rating Scale—IV* | BAARS-IV | Barkley [79] | *Selbsteinschätzung emotionaler Kompetenzen / Emotion Regulation Skills Questionnaire* | SEK-27 / ERSQ | Berking and Znoj [92] / Grant et al. [93] |
| *Conners`Adult ADHD Rating Scales* | CAARS | Conners et al. [80] | *Skalen zum Erleben von Emotionen (Experience of Emotions Scale)* | SEE (EES) | Behr and Becker [94] |
| *Adult Interview; Childhood ADHD Symptom Scale–Self-Report, and Current ADHD Symptoms Scale* | | Barkley and Murphy [81] | *Deficient Emotional Self-Regulation questionnaire (from Current Behaviour Scale)* | DESR | Barkley [95] |
| *Wender–Reimherr Adult Attention Deficit Disorder Scale* | WRAADDS | Wender [82] | *Reactivity, Intensity, Polarity, and Stability questionnaire* | RIPoSt-40 | Brancati et al. [96] |
| *Wender Utah Rating Scale* | WURS | Ward et al. [83] | *Wender–Reimherr Adult ADHD–Emotion Dysregulation Scale* | WRAADDS-EDS | Wender [82] |
| *ADHD–Self Report* | ADHD-SR | Rösler et al. [84] | *Behavior Rating Inventory of Executive Function, Adult Version—Emotional Control Subescale* | BRIEF-A-EC | Roth et al. [97] |
| *ADHD Clinical Diagnostic Scale Version 1.2* | ACDS v1.2 | Adler and Cohen [85], Adler and Spencer [86] | *Adult ADHD Investigator Rating Scale–Emotional Dyscontrol subscale* | AISRS-ED | Silverstein et al. [87] |
| *Adult ADHD Investigator Rating Scale.* | AISRS | Silverstein et al. [87] | *The Comprehensive Emotion Regulation Inventory* | CERI | Thorell et al. [53] |

increases in Impulse Control Difficulties [$p = 0.005$] and Lack of Emotional Clarity [$p = 0.014$] subscales.

## Neurological and psychophysiological activity related to emotion dysregulation in adult ADHD

Matthies et al. [69] measured psychophysiological variables in a group of adults with ADHD during a sadness induction experimental task. The participants were divided into two groups, each of which received specific instructions to carry out the task based on emotion regulation strategies: acceptance of emotion (ACC) and suppression of emotion (SUP). As a dependent variable, they measured on three occasions throughout the experiment the level of sadness that the subject was experiencing (T1 = before the video, T2 = immediately after the video, T3 = after viewing landscapes). The results indicated that the participants in both conditions did not differ significantly in the level of feeling overwhelmed by emotion in T1 and T2, but in T3 [SUP: 26.61 / 26.97, ACC: 8.11 / 10.71, $t_{1,34} = 2.71$, $p < 0.050$], pointing out a slower recovery of emotion in the suppression condition. Additionally, they found a paradoxical decrease in the activation of the sympathetic autonomic nervous system.

Regarding brain activity, four studies that were conducted with adults diagnosed with ADHD were reviewed. In the first study, Shushakova et al. [52] analyzed emotion regulation strategies in adults with and without ADHD using Event Related Potentials (ERP). They

observed a greater amplitude of the Late Positive Potential (LPP) in the ADHD group than in the control group [$F_{1,69}$ = 3.908, $p$ = 0.052, $d$ = 0.47]. Likewise, the group effect was significant, showing that the frontal amplitudes of the LPPs were higher in ADHD adults without medication compared to the control group [$F_{1,63}$ = 7.083, $p$ = 0.010, $d$ = 0.69]. In ADHD cases, the habitual use of reappraisal was associated with smaller LPP amplitudes in the SUP condition [$r$ = −0.44, $p$ = 0.005]. In a second study also using ERPs, Shushakova et al. [70] investigated attentional bias towards emotional stimuli in adults with ADHD using the dote-probe paradigm. They focused on the N2pc component that has classically been related to emotion dysregulation in healthy people [70]. The ANOVA of omission rates revealed a significant group x emotion interaction [$F_{1,78}$ = 7.102, $p$ = 0.009]. Post hoc $t$ Tests indicated that ADHD patients made significantly fewer omissions in the trials with positive stimuli than in the negative trials [$t_{38}$ = 2.104, $p$ = 0.042, $d$ = 0.322], while the controls showed a similar pattern of omissions. The ANOVA of accuracy revealed a significant group x congruence interaction [$F_{1,78}$ = 6.358, $p$ = 0.014], ADHD being less accurate than the controls in the congruent condition [$t_{78}$ = 2.748, $p$ = 0.007, $d$ = 0.613]. Different $t$ tests showed that in the ADHD group the N2pc component was significant for both conditions, positive [$t_{38}$ = 2.211, $p$ = 0.033] and negative [$t_{38}$ = 2.048, $p$ = 0.048], while for the controls it was only significant in the positive [$t_{40}$ = 2.475, $p$ = 0.01] condition. Overall, the amplitude of the P1 component was smaller in the ADHD group compared to the controls [$F_{1,78}$ = 9.488, $p$ = 0.003]. Regarding the clinical correlates, the authors observed a positive correlation between the SEK-27 scores and the N2pc amplitude [$r$ = 0.35, $p$ = 0.033], which differed significantly from the control group [$r$ = -0.106, $p$ = .511; $z$ = -1,999, $p$ = 0.023].

Using functional magnetic resonance imaging (fMRI), Materna et al. [51] monitored brain activity while the participants (adults with ADHD and healthy controls) performed an emotional regulation task, consisting of "viewing" neutral or negative images or "reappraisal" of negative images. In the reappraisal condition, participants were instructed to reduce their negative emotional response by reinterpreting the image in a less negative way. They found a significant interaction effect when comparing neutral-negative conditions in ADHD with controls, resulting in three differential activity clusters in the dorsal and ventral anterior cingulated cortex, even after being corrected for multiple comparisons [$p$ <0.05].

Finally, Li et al. [54] analyzed the relationship between resting-state EEG and daily executive functioning in adults with ADHD with high intelligence quotient (IQ). These authors found an association between relative beta power in central region and emotional control scores for ADHD [$r$ = 0.464, $p$ = 0.027], but did not in the control group [$r$ = 0.129, $p$ = 0.385].

## Emotion regulation outcomes in adult ADHD treatment

As far as we know just four studies have measured ER as outcome in interventions to date. Mitchell et al. [71] observed a group (ADHD-waiting list) x time (before-after the intervention) interaction, with an improvement after a randomized-controlled intervention based in Mindfulness training in the inattention symptoms [$p \leq 0.002$, $d \geq 1.09$], in hyperactivity-impulsivity symptoms [$p \leq 0.008$, $d \geq 0.75$], in functional impairment [$p \leq 0.003$, $d \geq 1.52$], in executive functions [$p \leq 0.005$, $d \geq 1.45$] and in emotion dysregulation [$p \leq 0.002$, $d \geq 1.27$]. Significant differences before-after treatment compared to the control group were found on the DERS Impulse Control Difficulties and Limited Access to Emotion Regulation Strategies subscales, both experiencing a significant reduction [$p$ = 0.002, $d$ = 1.60, and $p$ = 0.007, $d$ = 1.38, respectively]. However, Badoud et al. [72] did not obtain significant changes in any measure after a mentalization-based intervention without a control group, observing lower scores on the ASRSv1.1 and Emotion Reactivity Scale (ERS) subscales, which would indicate a reduction in symptoms, with small effect sizes.

Lastly, Reimherr et al. [73, 74] measured the effect on ED of atomoxetine and osmotic release oral system (OROS) methylphenidate in two double-blinded randomized-controlled trials in adults with ADHD. They found that both atomoxetine and OROS methylphenidate medications improved ED measured through the Wender–Reimherr Adult ADHD–Emotion Dysregulation Scale (WRAADDS-EDS) compared to a placebo [$F = 15.10$, $p = 0.001$, $d' = 0.66$; $F = 10.476$, $p = 0.002$, $d' = 0.70$; respectively].

## Discussion

The aim of this review was to unify available data related to ED in adults with ADHD. To date, few studies have been conducted with adult population diagnosed with ADHD on emotion dysregulation, which is altered in 34–70% of affected adults, according to the review carried out by Shaw et al. [31]. However, some preliminary outcomes suggests that not only inattention, hyperactivity, and impulsiveness, but also emotional dysregulation are core components of ADHD [13] (see also [12, 98, 99] for similar conclusions). Despite this, no significant differences in ED were found between people with ADHD and people with other disorders, such as borderline personality disorder (BPD) [58, 62], being a controversial point.

According to our first objective of analyzing ER features of people with ADHD, the studies reviewed suggest that when adults with or without ADHD are compared, in line with our first hypothesis (H1), the ADHD group have consistently lower ER scores than controls using different measures ($k = 10$) [51–54, 58–63]. Although a differential response pattern between ADHD and other disorders (examples) could not be identified, evidence may indicate that high emotional dysregulation scores are associated with greater socio-functional impairment as H1 proposed ($k = 3$) [13, 63, 66]. Such emotional impairment could be the reason why health problems are usually observed in adults with ADHD, for instance those related to accidents and substance misuse, psychiatric comorbidity, poor academic and occupational performance [6, 100, 101], as well as in their personal relationships, self-esteem, and daily activities [100].

In terms of the second hypothesis (H2), another consistent finding is the habitual use of non-adaptive ER strategies in adults with ADHD ($k = 4$). For example, Rüfenacht et al. [63] observed an ADHD trend to blame themselves, catastrophize, blame others, and ruminate. Studies using ERQ consistently report a greater use of the suppression strategy and a lower use of reappraisal in the ADHD group compared to controls [51, 52, 70]. However, Matthies et al. [69] found that when using the suppression of emotion, a slower return to the baseline occurs in ADHD adults, not being so when using acceptance. This could indicate that despite being the most widely used strategy in this population, its implementation causes greater emotional distress by prolonging the time in which negative emotions are experienced, maintaining the use of other non-adaptive strategies (rumination, catastrophizing, etc.) and, consequently, of emotion dysregulation, in a kind of vicious circle. Nevertheless, it could also indicate the possibility of improving the emotional response by training on strategies used to regulate emotions, as it has been observed that when ADHD symptoms improve, so does emotional dysregulation [31, 71]. Likewise, the application of specific training in ER in children with ADHD may improve their ER skills [102]. It also seems that the usual pharmacological treatments for ADHD improve the symptoms of emotional dysregulation [73, 74], reducing their emotional discomfort.

Regarding our second objective of analyzing ADHD symptoms and its relationship with ED, we conclude that it is not possible to establish a clear distinction between ADHD subtypes based on performance on ER measurement instruments with the available evidence. However, it has been observed that the combined subtype tends to load more ED [60] and that the

greater the severity of ADHD symptoms, the greater the emotion dysregulation [63]. These data are in line with the results obtained in other studies carried out with children and adolescents with ADHD [103]. In contrast, Hirsch et al. [13] considered the presence of high negative affect and failure to implement adaptive ER strategies as a distinctive feature of adult ADHD. Similarly, Edel et al. [65] identified having parents with ADHD symptoms as a risk factor of ED, with maternal symptoms having the greatest impact. These last results are supported by a subsequent study carried out by Mazursky-Horowitz et al. [104], in which they found that the ED in mothers mediated the relationship between maternal ADHD symptoms and severe parenting responses to the expression of negative emotions by adolescents.

Concerning intervention programs to improve ED in people with ADHD, preliminary data points to the effectiveness of Mindfulness training as an effective intervention for this purpose [71]. However, the results of the mentalization-based intervention are not significant, despite appreciating a trend towards improvement in symptoms [72]. More studies of treatments that consider emotional variables are needed. Additionally, it is interesting to highlight the results obtained by Mitchell et al. [71] regarding the improvement in EF measured with questionnaires, but not with lab task. According to Barkley [10], daily EF and ER correlate when measured with rating scales, but this is not what the lab tasks showed, as he has challenged. Silverstein et al. [66] obtained evidence that supports these last results.

Finally, our third objective focused on identifying characteristics of brain functioning in adults with ADHD. Using fMR, Materna et al. [51] did not find differences in the brain activation of people with and without ADHD in the condition of explicit emotional regulation when participants were instructed to reappraise the image in a less negative way. However, given the condition of view, where they had to attend to the image without performing avoidance behaviors, authors found significant differences in the cortical activation of both groups, with the dorsal and ventral anterior cingulated cortex being highly activated in the ADHD group compared to control group. Authors suggested that these differences are due to the implementation of implicit regulatory strategies. However, as we have pointed out previously, people with ADHD often use emotional suppression strategies rather than reappraisal ones. In these cases, therapeutic training in the use of more adaptive regulatory strategies, such as reappraisal, could improve their emotional competence, as noted above.

For their part, Shushakova et al. [52, 70] focused on evoked potentials, finding that the amplitude of the LPP in the central parietal and frontal areas, as well as the amplitude of the N2pc (related to attentional bias) were associated with the severity of the ADHD symptoms. The N2pc component confirms the attention bias that people with ADHD manifest towards emotional stimuli. The greater amplitude of the frontal LPP suggests a greater cognitive effort to regulate their emotions and would support the hypothesis of emotional hyper-reactivity in ADHD when they process aversive stimuli. This type of activity would fit with the difficulties of top-down regulation, that is, the modulating effect that EF would have on the emotional reaction at the base through frontal networks. According to Barkley [12], this fact would reflect that dysregulation in ADHD is a top-down problem rather than a bottom-up problem, and would be independent of the IQ, since adults with ADHD with high IQ also showed impairments in daily EF along with aberrant resting-state EEG patterns relative to controls [54].

Overall, anomalous activation patterns are observed in adults with ADHD compared to adults without ADHD. It is well-known that the anterior cingulate cortex integrates attentional, mnemonic, and emotional information and activates the dlPFC, which promotes decision-making based on emotional and objective information [19]. Abnormal activation patterns seen in ADHD may interfere with emotional processing from the beginning, biasing attention toward emotional salient stimuli [52, 70, 105, 106]. This would limit ER and executive processes carried out by the dlPFC, due to the lack of relevant emotional and contextual

information. In this line, one possible strategy of interventions that could be used to improve these impaired patterns in ADHD patients may focus on attention processes such as Mindfulness, given certain preliminary outcomes [71, 107]. This idea finds support in Bailey and Jones' integrated model of regulation [108]. According to this model, the development of ER would rely on the establishment of essential components, such as attention and inhibition mechanisms that are affected in ADHD from early childhood.

Some limitations need to be highlighted. The studies found that meet the selection criteria are scarce and very heterogeneous both in aims and in sample features, mainly in variables as relevant as sex, medication status and comorbidity. As noted above, not taking gender into account, as well as the presence of comorbidities such as BPD, could overestimate ED scores differences between ADHD and controls. In contrast, ADHD drugs could mask them, as they tend to reduce ED severity symptoms [73, 74]. The lack of unifying criteria for the assessment of both ADHD and ED makes it difficult to compare studies, as well as to draw generalizable conclusions. Furthermore, recent research suggests the need to add performance tests to ADHD diagnosis to avoid malingering [109]. Symptom-based tests have been shown to be less effective in detecting it, but they are the main source of diagnosis [109]. This fact of high heterogeneity reduces the potential for future meta-analyses on this topic unless they are limited to a small subset of comparable studies.

To conclude, our findings highlight the emotional side of ADHD in adults and the convenience of new research on the topic. Although emotional dysregulation is present in ADHD, its presence in other disorders does not justify that ED is a core symptom of ADHD. The establishment of flexible, widely agreed guidelines for diagnosis, including performance and symptom-based tests, and psycho-educational interventions focused on improving the impact of emotional dysregulation is needed. Future studies are necessary to jointly analyze the results obtained with different measures of ER, including ER strategies. Other aspects such as emotional recognition and processing, which are a prerequisite for optimal ER [42], need more research in adults with ADHD as well. In addition, the impact of emotional dysregulation on adult ADHD needs to be analyzed in terms of sex differences, comorbidity, and medication treatments.

## Supporting information

**S1 Table. Quality analysis of selected papers.**
(DOCX)

**S2 Table. Demographic information of selected papers.**
(DOCX)

**S1 File. Dataset.**
(XLSX)

**S1 Checklist. PRISMA 2020 checklist.**
(DOCX)

## Author Contributions

**Conceptualization:** Ana-María Soler-Gutiérrez, Juan-Carlos Pérez-González, Julia Mayas.

**Data curation:** Ana-María Soler-Gutiérrez.

**Formal analysis:** Ana-María Soler-Gutiérrez.

**Funding acquisition:** Ana-María Soler-Gutiérrez.

**Supervision:** Juan-Carlos Pérez-González, Julia Mayas.

**Visualization:** Ana-María Soler-Gutiérrez.

**Writing – original draft:** Ana-María Soler-Gutiérrez.

**Writing – review & editing:** Ana-María Soler-Gutiérrez, Juan-Carlos Pérez-González, Julia Mayas.

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
