## [Decision Letter · Decision Letter 0]

13 Oct 2022

PONE-D-22-18919Evidence of emotion dysregulation as a core symptom of adult ADHD: a systematic reviewPLOS ONE

Dear Dr. Julia Mayas,

Thank you for submitting your manuscript to PLOS ONE. After careful consideration, we feel that it has merit but does not fully meet PLOS ONE’s publication criteria as it currently stands. Therefore, we invite you to submit a revised version of the manuscript that addresses the points raised during the review process.

We look forward to receiving your revised manuscript.

Kind regards,

Yoshiyuki Tachibana

Academic Editor

PLOS ONE

Journal Requirements:

This work was supported by Universidad Nacional de Educación a Distancia (UNED) through an FPI-UNED grant given to A.M.S.G.

Additional Editor Comments:

Dear Dr. Yvonne Groen:

Manuscript ID PONE-D-22-18919 entitled "Evidence of emotion dysregulation as a core symptom of adult ADHD: a systematic review" which you submitted to PLOS ONE, has been reviewed favorably and minor revisions have been requested. I invite you to respond to the comments appended below and revise your manuscript.

Reviewers' comments:

Reviewer's Responses to Questions

**Comments to the Author**

1. Is the manuscript technically sound, and do the data support the conclusions?

Reviewer #1: Yes

2. Has the statistical analysis been performed appropriately and rigorously? 

Reviewer #1: N/A

3. Have the authors made all data underlying the findings in their manuscript fully available?

Reviewer #1: Yes

4. Is the manuscript presented in an intelligible fashion and written in standard English?

Reviewer #1: Yes

5. Review Comments to the Author

Reviewer #1: I would like to congratulate the authors on this interesting and clearly written systematic literature review. Emotion dysregulation in adults with ADHD is a timely topic and a review would be a welcome contribution to the existing literature. The introduction provides a solid background of the field with recent references. The results are well organized and supported by exhaustive tables. The discussion section summarizes the main findings and conclusions, which are related back to the set hypotheses.

My main point of feedback is about the discussion section: it could benefit from a more critical discussion of the limitations and presentation of a research agenda. For example, the review identified several factors that need to be taken into account when studying emotion regulation in ADHD, such as sex differences, medication use, different types of emotion regulation strategies, overlap with other disorders, etc. It would be very helpful to field if the most important factors would be discussed, so to foster high quality research in this field. Perhaps the authors could think of a research agenda? One important point that needs consideration is the specificity of emotion regulation problems for ADHD; it is stated (p.21, line 337) that emotion dysregulation are core components of ADHD. However, this conclusion may be too preliminary, since also other disorders are characterized by these impairments, e.g. autism, anxiety, depression. In order to conclude that emotion dysregulation is part of the disorder, more studies need to be carried out to determine the specificity for ADHD and also about the causes of these problems. For example, it is mentioned that these problems are the result of top-down processes and that emotion regulation improves when top-down processes improve. Emotion dysregulation may therefore be secondary to cognitive dysregulation. The authors might be interested in a recent model of regulation for applied settings by Bailey & Jones (2019) https://doi.org/10.1007/s10567-019-00288-y, showing that emotion regulation needs to be integrated in research with cognitive as well as social regulation. Emotion regulation is just one aspect of self-regulation in humans and may covary with the other domains, in the case of ADHD most likely the cognitive domain. Furthermore, some methodological limitations could be discussed, such as the use of self-reports which could be less reliable in adults with ADHD because of over- or underestimation of their own symptoms on retrospective questionnaires, e.g. Wallace et al., 2019, https://doi.org/10.1037/pas0000659 . It may be necessary to use more objective or ecological momentary assessments for emotion regulation in future studies.

Below I formulated several minor errors and irregularities I found in the paper:

Abstract: nuclear symptom is a strange wording -> core symptom

Whole paper: sometimes the word emotional (dys)regulation is used -> use consistently emotion (dys)regulation; sometimes the word subjects is used to describe the participants, which is a disrespectful term for the participants and should be avoided according to APA style.

Introduction: the aims could be written more concise by coupling aim and hypotheses, less enters need to be used.

Method: p.7 it says that participants over age 18 were included, but the flow-chart says over 16 – which is the correct? In the procedure section it needs mentioning why no meta-analysis was conducted – this only comes back in the discussion. Tables 1&2 could be added to the supplements, because they only provides background information.

Results: p.15 line 206 was the ER score consistently lower? How many studies did not find this effect? P.16 line 216 variance of what? Regarding the treatment studies: clear up if the outcome measures were blinded/ who the rater was.

Discussion: p.21 line 335 it says that 34-70% of adults with ADHD are affected by emotion dysregulation, but it is unclear whether this stems from the review results or just one study. It is a bit confusing in this first paragraph whether these are conclusions from the review or not. When discussing the findings it would be helpful to mention the number/percentage of studies that found positive or negative results to get an idea of the consistency of the findings. I am not sure whether the conclusion that emotion dysregulation is a core component of ADHD is sufficiently supported by the review and current research (see above). A final concluding paragraph would be helpful.

6. PLOS authors have the option to publish the peer review history of their article (what does this mean?). If published, this will include your full peer review and any attached files.

Reviewer #1: **Yes: **Dr. Yvonne Groen

---

## [Author Response · Author response to Decision Letter 0]

24 Nov 2022

Responses to Journal requirements:

We have check that all files meet PLOS ONE’s requirements.

This work was supported by Universidad Nacional de Educación a Distancia (UNED) through an FPI-UNED grant given to A.M.S.G.

In our reviewed submission we have stated that the funders had no role in study design, data collection and analysis, decision to publish, or preparation of the manuscript, as indicated in our cover letter. 

We have attached our research dataset as a Supporting Information excel file. We state in our cover letter that our dataset is fully available.

Correspondence author has validated her ORCID iD in Editorial Manager.

We have included captions for our Supporting Information files at the end of our manuscript and update in-text citations accordingly. 

We have included two news references [108, 109] following the recommendations of our reviewer.

Responses to the Reviewer's comments:

Reviewer #1: I would like to congratulate the authors on this interesting and clearly written systematic literature review. Emotion dysregulation in adults with ADHD is a timely topic and a review would be a welcome contribution to the existing literature. The introduction provides a solid background of the field with recent references. The results are well organized and supported by exhaustive tables. The discussion section summarizes the main findings and conclusions, which are related back to the set hypotheses.

Thank you very much for your kind feedback. We greatly appreciate the suggested improvements and invitation to submit a revised version of our manuscript.

My main point of feedback is about the discussion section: it could benefit from a more critical discussion of the limitations and presentation of a research agenda. For example, the review identified several factors that need to be taken into account when studying emotion regulation in ADHD, such as sex differences, medication use, different types of emotion regulation strategies, overlap with other disorders, etc. It would be very helpful to field if the most important factors would be discussed, so to foster high quality research in this field. Perhaps the authors could think of a research agenda? 

Firstly, thank you for bringing up this point. We have followed this recommendation discussing three main factors that future research need to consider: sex differences, medication status and pre-existing comorbidities, as they could get conflicting outcomes.

One important point that needs consideration is the specificity of emotion regulation problems for ADHD; it is stated (p.21, line 337) that emotion dysregulation are core components of ADHD. However, this conclusion may be too preliminary, since also other disorders are characterized by these impairments, e.g. autism, anxiety, depression. In order to conclude that emotion dysregulation is part of the disorder, more studies need to be carried out to determine the specificity for ADHD and also about the causes of these problems. For example, it is mentioned that these problems are the result of top-down processes and that emotion regulation improves when top-down processes improve. Emotion dysregulation may therefore be secondary to cognitive dysregulation. The authors might be interested in a recent model of regulation for applied settings by Bailey & Jones (2019) https://doi.org/10.1007/s10567-019-00288-y, showing that emotion regulation needs to be integrated in research with cognitive as well as social regulation. Emotion regulation is just one aspect of self-regulation in humans and may covary with the other domains, in the case of ADHD most likely the cognitive domain. 

We totally agree with this suggestion. The evidence obtained to date does not allow us to ensure that ED is an ADHD core symptom, so we have made it clear. Thank you for such a useful reference, so it fits with our research and help to explain our conclusions. That is why we have included it.

Furthermore, some methodological limitations could be discussed, such as the use of self-reports which could be less reliable in adults with ADHD because of over- or underestimation of their own symptoms on retrospective questionnaires, e.g. Wallace et al., 2019, https://doi.org/10.1037/pas0000659 . It may be necessary to use more objective or ecological momentary assessments for emotion regulation in future studies.

Certainly, the diagnostic process for ADHD needs to include more objective and reliable tests. We have taken into account the suggested reference to improve our discussion on this issue. 

Below I formulated several minor errors and irregularities I found in the paper:

Abstract: nuclear symptom is a strange wording -> core symptom

We have changed nuclear for the better one core

Whole paper: sometimes the word emotional (dys)regulation is used -> use consistently emotion (dys)regulation; 

We made changes needed to use consistently emotion dysregulation. However, in some cases was not appropriate since reference is made to the global process of emotion regulation, so we maintain the term emotion regulation in these occasions. 

sometimes the word subjects is used to describe the participants, which is a disrespectful term for the participants and should be avoided according to APA style.

Thank you for letting us know about this issue. We have replaced subjects for participants.

Introduction: the aims could be written more concise by coupling aim and hypotheses, less enters need to be used.

We have reformulated this point to make it more concise.

Method: p.7 it says that participants over age 18 were included, but the flow-chart says over 16 – which is the correct? In the procedure section it needs mentioning why no meta-analysis was conducted – this only comes back in the discussion. 

Certainly, it was an error in age. We have corrected it in the flowchart. In addition, we have clarified the reason for not carrying out meta-analyses

Tables 1&2 could be added to the supplements, because they only provides background information.

Following this suggestion, we have added tables 1 and 2 as supporting information and made the necessary adjustments to the main text.

Results: p.15 line 206 was the ER score consistently lower? How many studies did not find this effect? P.16 line 216 variance of what? Regarding the treatment studies: clear up if the outcome measures were blinded/ who the rater was.

We have indicated in the text the number of studies that found the commented results, also in discussion section. Regarding treatments, we have clarified if they were controlled and/or blinded, but in three of them we are not clear the measures of ED are self-reported or observed by the clinician. 

Discussion: p.21 line 335 it says that 34-70% of adults with ADHD are affected by emotion dysregulation, but it is unclear whether this stems from the review results or just one study. It is a bit confusing in this first paragraph whether these are conclusions from the review or not. When discussing the findings it would be helpful to mention the number/percentage of studies that found positive or negative results to get an idea of the consistency of the findings. I am not sure whether the conclusion that emotion dysregulation is a core component of ADHD is sufficiently supported by the review and current research (see above). 

We have reformulated the first discussion paragraph to clarify this issue, since it is a conclusion of other authors. Also explaining the controversy of whether ED is a core symptom of ADHD. Throughout the section, we have indicated the number of articles that obtain the mentioned results.

A final concluding paragraph would be helpful.

Finally, we have made some changes in the last part of the discussion, writing the last paragraph as a conclusion.

6. PLOS authors have the option to publish the peer review history of their article (what does this mean?). If published, this will include your full peer review and any attached files.

Do you want your identity to be public for this peer review? For information about this choice, including consent withdrawal, please see our Privacy Policy.

Reviewer #1: Yes: Dr. Yvonne Groen

---

## [Editor Report · Decision Letter 1]

21 Dec 2022

Evidence of emotion dysregulation as a core symptom of adult ADHD: a systematic review

PONE-D-22-18919R1

Dear Dr. Julia Mayas,

We’re pleased to inform you that your manuscript has been judged scientifically suitable for publication and will be formally accepted for publication once it meets all outstanding technical requirements.

Kind regards,

Yoshiyuki Tachibana

Academic Editor

PLOS ONE
---

## [Editor Report · Acceptance letter]

27 Dec 2022

PONE-D-22-18919R1 

Evidence of emotion dysregulation as a core symptom of adult ADHD: a systematic review 

Dear Dr. Mayas:

I'm pleased to inform you that your manuscript has been deemed suitable for publication in PLOS ONE. Congratulations! Your manuscript is now with our production department. 

Kind regards, 

on behalf of

Dr. Yoshiyuki Tachibana 

Academic Editor

PLOS ONE